

# The role of dispersal for shaping phylogeographical structure of flightless beetles from the Andes

Sofia I. Muñoz-Tobar[1,2] and Michael S. Caterino[1]

[1] Department of Plant & Environmental Sciences, Clemson University, Clemson, SC, USA
[2] Escuela de Ciencias Biológicas, Pontificia Universidad Católica del Ecuador, Quito, Pichincha, Ecuador

## ABSTRACT

**Background:** Páramo is a tropical alpine ecosystem present in the northern Andes. Its patchy distribution imposes limits and barriers to specialist inhabitants. We aim to assess the effects of this habitat distribution on divergence across two independently flightless ground beetle lineages, in the genera *Dyscolus* and *Dercylus*.
**Methods:** One nuclear and one mitochondrial gene from 110 individuals from 10 sites across the two lineages were sequenced and analyzed using a combination of phylogenetics, population genetic analyses, and niche modeling methods.
**Results:** The two lineages show different degrees of population subdivision. Low levels of gene flow were found in *Dyscolus alpinus*, where one dominant haplotype is found in four out of the six populations analyzed for both molecular markers. However, complete population isolation was revealed in species of the genus *Dercylus*, where high levels of differentiation exist at species and population level for both genes. Maximum entropy models of species in the *Dercylus* lineage show overlapping distributions. Still, species distributions appear to be restricted to small areas across the Andes.
**Conclusion:** Even though both beetle lineages are flightless, the dispersal ability of each beetle lineage appears to influence the genetic diversity across fragmented páramo populations, where *Dyscolus alpinus* appears to be a better disperser than species in the genus *Dercylus*.

## INTRODUCTION

The distribution of a species is determined by environmental conditions, ecological interactions of species, and dispersal dynamics. In particular, the dispersal ability of species plays an important role in determining species ranges and biogeographical patterns (*Lester et al., 2007*). The evolution of wings in insects is considered a major factor in their success and diversity (*Stone & French, 2003*; *Nicholson, Ross & Mayhew, 2014*), since it has allowed them to avoid predators, capture prey, and disperse (*Stone & French, 2003*). Yet, the secondary loss and reduction of wings is recorded across many insect lineages (*Wagner & Liebherr, 1992*), and wing length polymorphisms in insects, from reduction to complete loss of wings and wing muscles, appear to have strongly influenced dispersal and

Corresponding author
Sofia I. Muñoz-Tobar,
munoztobarsofia@gmail.com

biogeographical patterns in these lineages (*Gutiérrez & Menéndez, 1997*; *Mcculloch, Wallis & Waters, 2017*; *Ikeda, Nishikawa & Sota, 2012*).

Flightlessness has been recorded in insect faunas in stable habitats such as oceanic islands and caves (*Darwin, 1859*; *Wagner & Liebherr, 1992*), as well as in species that live in harsh environments, like polar and alpine regions (*Somme & Block, 1991*). The mechanism of selection against wings in beetles was hypothesized by *Darwin (1859*; pp. 135–136*)*, based on the observation of beetles from an oceanic island, where flight could cause displacement due to high wind in open areas. It has also been suggested that maintaining functional wings is energetically expensive (*Darlington, 1943*; *den Boer et al., 1979*). Studies across beetle species show that most winged species appear to have a wider geographical distribution than brachypterous species (*Gutiérrez & Menéndez, 1997*; *Ikeda, Nishikawa & Sota, 2012*). Therefore, the loss of flight capabilities is predicted to promote allopatric speciation due to limited dispersal power (*Ikeda, Nishikawa & Sota, 2012*). Still, beetle communities in montane regions are often composed of winged species (macropterous), species with reduction of wings (brachypterous), species with very small wings (micropterous), as well as wing–polymorphic species that exhibit a range in the length of wings (*Nilsson, Petterson & Lemdahl, 1993*; *Moret, 2005*).

Flightlessness in carabid beetles has been the focus of several studies, some of which support the idea that wing loss promoted genetic isolation across populations (*Sota & Nagata, 2008*; *Homburg et al., 2013*). However, lack of genetic structure has also been reported for other flightless ground beetles, where populations appear to maintain low levels of gene flow (*Chatzimanolis & Caterino, 2007*). In a comparative study across five carabid lineages in the eastern United States, some winged and wingless forms were found to have comparable levels of genetic heterogeneity (*Liebherr, 1988*). This broad range of observations suggests that population differentiation results from a combination of factors, including dispersal capability, but also discontinuity of the habitat, resource availability, and population sizes (*Liebherr, 1988*; *Vogler, 2012*).

The higher elevations of the northern Andes host a unique plant community known as páramo. Limited to elevations above 3,000 m, páramo is patchily distributed among the many higher peaks and volcanos in the range. Páramo contains a high number of endemic species, which are mostly are restricted to small elevational ranges (above 2,800 m; *Luteyn, 1999*; *Sklenář, Dušková & Balslev, 2011*; *Madriñán, Cortés & Richardson, 2013*; *Sklenář, Hedberg & Cleef, 2014*). Habitat loss due to various anthropogenic impacts is a major concern for the conservation of this ecosystem (*Hofstede et al., 2002*), but little consensus has emerged on the effects of habitat isolation on diversification of this region's biota. Phylogenetic studies of high elevation species from the Andes have been few and mostly centered on plants (*Madriñán, Cortés & Richardson, 2013*; *Hughes & Atchison, 2015*; *Gómez-Gutiérrez et al., 2017*). These show mostly high diversification rates, and in some instances, signs of the contraction and expansion of the páramo during Quaternary glaciations affecting some plant lineages (*Gómez-Gutiérrez et al., 2017*). For the few vertebrate species studied, varied patterns of differentiation are observed (*Páez-Moscoso & Guayasamin, 2012*; *Rodríguez Saltos & Bonaccorso, 2016*), and few insects or other invertebrate species have been examined (*Hines, 2008*; *Elias et al., 2009*; *Polato et al., 2018*).

Analyses of aquatic insects from páramo show these insect lineages possess narrow thermal tolerance ranges, which appear to promote high speciation rates (*Polato et al., 2018*).

In this study we focused on two flightless ground beetle (Coleoptera: Carabidae) lineages from páramo. Most of the ground beetles present in páramo are micropterous (76% of species), with only few macropterous and brachypterous species reported (*Moret, 2005*). To examine the effect of fragmentation across páramo patches we examined two distinct lineages. The first, *Dyscolus alpinus Chaudoir*, 1878 (Coleoptera, Carabidae, Harpalinae, Platynini), is a micropterous species present in the northern and central portion of the Ecuadorian Andes, between 2,750 and 4,200 m, in páramo, subpáramo, and montane forest (*Moret, 2005*). Members of the genus *Dyscolus Dejean, 1831* are mainly neotropical, from Mexico to Argentina, with some representatives present in the southern United States (*Moret, 2005*; *Bousquet, 2012*). Of 320 total species in the genus *Dyscolus*, 89 species are known from Ecuadorian páramo (2,750–4,200 m). The second beetle lineage assessed comprises multiple species in the genus *Dercylus Laporte de Castelnau, 1832* (Coleoptera, Carabidae, Harpalinae, Licinini). *Dercylus* is a new world genus with 35 species (*Bousquet, 2012*), including five species from the Ecuadorian Andes (*Moret & Bousquet, 1995*; *Moret, 2005*). All Ecuadorian species belong to the subgenus *Linodercylus Kuntzen, 1912*, all of which are micropterous and occurring in montane forest and páramo between 2,700 and 4,200 m. Species sampled for analyses include three described species: *Dercylus orbiculatus Moret & Bousquet, 1995*, *Dercylus praepilatus Moret & Bousquet, 1995*, and *Dercylus cordicollis* (*Chaudoir, 1883*), and two as-yet unidentified (probably undescribed) species from the Ecuadorian páramo.

The aim of this study is to analyze the effects of montane isolation in these two flightless beetle lineages from páramo, and to ask how their phylogenetic structure and population connectivity have been affected by potential geographical barriers, their dispersal abilities, and their ages of divergence. We analyzed two molecular markers, cytochrome c oxidase subunit I (COI) and carbamoyl phosphate synthetase 2 (CAD), alone and combined, to determine phylogenetic relationships among individuals for each lineage, population genetic diversity and structure, and approximate ages of each clade with the use of molecular clocks. In addition to the analyses of the molecular data, geographic information was used to generate models of species distribution as a way of assessing habitat continuity.

## MATERIALS AND METHODS

### Study area

The study area encompassed the Ecuadorian Andes, where all the sampling took place above 3,100 m in the páramo ecosystem. The selection of sites was based on previous records of occurrence of the target taxa, and potential geographical barriers. Some of these potential barriers include: dry valleys, rivers, faults, and the east/west split of the Andes mountain range (*Krabbe, 2008*; *Guayasamin et al., 2010*; *Quintana et al., 2017*).

In particular, the Pastaza, Chanchan, and Mira rivers, and their corresponding valleys, appear to limit the distributions of some bird, amphibian, and plant species (*Krabbe, 2008*; *Guayasamin et al., 2010*; *Quintana et al., 2017*). The Pallatanga fault, a prominent NE–SW

strike–slip fault crossing the western cordillera (*Baize et al., 2015*), may also affect lineage dispersal. The most prominent geographic divide considered in this study was the split of the Ecuadorian Andes into east and west cordilleras, associated with limits of distribution in several vertebrate and plant species (*Chaves et al., 2007*; *Guayasamin et al., 2010*; *Sklenář, Dušková & Balslev, 2011*).

## Sample collection

Ground beetle specimens were collected at eight sites along the Ecuadorian Andes, between 3,100 and 4,000 m in national parks during two field seasons in the summers of 2015 and 2016 (Fig. 1; Table 1). Permissions necessary for the collection of samples were previously obtained through the Ministry of Environment of Ecuador (MAE), permit number MAE–DNG–ARGG–CM–2014–004). *Dyscolus alpinus* was found in six out of the eight sites sampled (Fig. 1A), while *Dercylus* specimens were collected in five sites (Fig. 1B). Adult beetles were collected into 100% EtOH using three methods: manual collection (underneath rocks and on vegetation), pitfall trapping, using two 100 m transects (one trap every 10 m), where traps ran for 2–3 days; and through the collection of leaf litter samples from the páramo floor. Three samples of leaf litter were collected per site, and were processed in the lab using Berlese funnels. Ground beetles were identified using *Moret's (2005)* taxonomic keys. An additional sample was provided and identified by Pierre Moret as *Dercylus cordicollis* (SIMT265) from Cotacachi, this sample was included for the analysis of *Dercylus cordicollis* populations. Outgroup taxa were chosen based on availability of fresh material and taxa considered closely related carabid lineage within the Harpalinae. Outgroup taxon sampling included 12 taxa: two were collected in the Ecuadorian páramo; six taxa were collected in the southeastern United States; and four additional samples were retrieved from GenBank (KR604910, Table S1). Species in the genera *Platynus*, *Agonum*, and *Incagonum* were used as outgroups in the analyses of *Dyscolus*. *Amara*, *Oodes*, *Dicaelus*, *Sarticus*, *Notonomus*, and *Loxodactylus* were used as outgroups in the analyses of *Dercylus* (Table S1). Voucher specimens will be deposited in the Museo de Zoología de la Pontificia Universidad Católica del Ecuador (QCAZ) after the study is completed.

## DNA extraction, sequencing, and alignment

Genomic DNA was extracted from 110 specimens (with an average of 10 individuals per site, per species) using GeneJet's Genomic DNA Purification Kit (Thermo Fisher Scientific, Vilnius, Lithuania) following the manufacturer's instructions. A fragment of the mitochondrial cytochrome c oxidase subunit I (COI) gene was amplified using the primers TL2–N–3014 (5′–TCCAATGCACTAATCTGCCATATTA–3′ and C1–J–2183 (5′–CAACATTTATTTTGATTTTTTGG–3′; *Simon et al., 1994*) and the amplification profile described in *Caterino & Tishechkin (2015)*. One nuclear protein-coding gene, CAD, was amplified using the primers CD439F (5′–TTCAGTGTACARTTYCAYCCHGARCAYAC–3′) and CD688R (5′–TGTATACCTAGAGGATCDACRTTYTCCATRTTRCA–3′). Heminested PCRs were done for difficult samples, using CD439F (5′–TTCAGTGTACARTT YCAYCCHGARCAYAC–3′) and CD1098R (5′–TTNGGNAGYTGNCCNCCCAT–3)
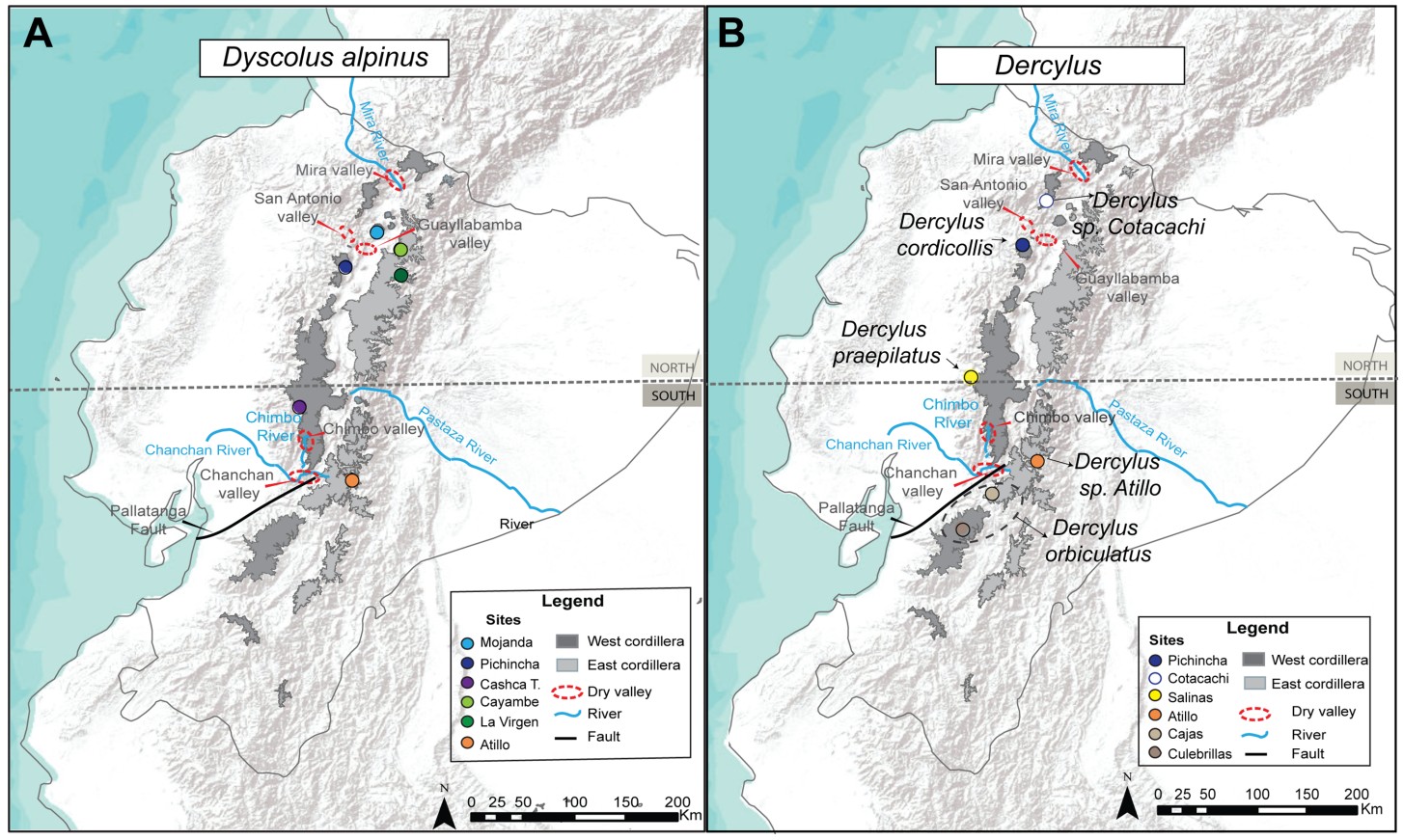

**Figure 1 Map of the Ecuadorian Andes displaying sites where samples of ground beetles were collected.** (A) *D. alpinus* and (B) species in the genus *Dercylus*. Major geographical barriers examined in the AMOVA are highlighted (east and west cordilleras, rivers, and dry valleys).

following *Wild & Maddison (2008)*. All PCRs were performed in 25 μl reactions, containing 17.5 μl water, 2.5 μl 10x buffer, 0.5 μl dNTPs, 0.75 μl MgCl2, 0.1 μl DreamTaq DNA Polymerase (Thermo Fisher Scientific, Vilnius, Lithuania) and one μl of each primer (five nm). PCR products were cleaned using ExoSAP–IT (USB/Affymetric, Santa Clara, CA, USA). Sequencing was done commercially by Macrogen USA, Inc. (Rockville, MD, USA).

DNA sequences were edited using Geneious R8 (Biomatters Ltd., Auckland, New Zealand) and aligned in MAFFT v.7 (*Katoh & Standley, 2013*) together with sequences from outgroups. Models of sequence evolution were tested using JModeltest 2.0 (*Darriba et al., 2012*) for each data set, where GTR + I + G model was in the 100% confidence interval for both COI and CAD data.

## Genetic diversity and population structure

The levels of genetic diversity were calculated using Arlequin 3.5.2.2 (*Excoffier & Lischer, 2010*) and DnaSP (*Rozas et al., 2003*) using the default settings. Groups analyzed were: (1) *Dyscolus alpinus*; (2) *Dercylus orbiculatus*; and (3) *Dercylus* species together (*Dercylus orbiculatus, Dercylus cordicollis, Dercylus praepilatus* and *Dercylus* sp. from Atillo and Cotacachi) in order to contrast inter– and intrapopulation variability. Parameters
**Table 1 Summary of the collecting sites from the Ecuadorian Andes, from which ground beetle species were collected for genetic analysis.**

| No. | Region | Site name | Latitude | Longitude | Elevation (m) | Col. Date |
|---|---|---|---|---|---|---|
| 1 | West Mountain range | Mojanda | N00°08.710′ | W78°16.753′ | 3,715 | July 12, 2016 |
| 2 | | Pichincha | S00°11.259′ | W78°32.432′ | 3,897 | June 22, 2016 |
| 3 | | Salinas | S1°24′11.03″ | W78°14.051′ | 3,604 | June 11, 2015 |
| 4 | | Cashca Totoras | S01°43.485′ | W78°57.183′ | 3,509 | June 13, 2015 |
| 5 | | Cajas | S02°47.020′ | W79°13.438′ | 3,956 | June 20, 2015 |
| 6 | East Mountain range | Cayambe | S00°02.101′ | W78°03.608′ | 3,743 | June 01, 2016 |
| 7 | | La Virgen | S00°18.477′ | W78°13.953′ | 3,694 | June 28, 2016 |
| 8 | | Atillo | S02°11.265′ | W78°31.2601′ | 3,501 | July 07, 2016 |
| 9 | | Culebrillas | S02°28.337′ | W78°53.719′ | 3,799 | June 15, 2015 |
| 10 | | Cotacachi | N00°19.79952′ | W78°20.80830′ | 3,757 | July 13, 2016 |

**Note:**
Data set compiled from 2015 to 2016.

estimated included: estimated haplotype diversity ($h$), nucleotide diversity ($\pi$), number of polymorphic sites ($S$), number of migrants per generation ($Nm$), and Tajima's D neutrality test statistic ($D$). $\phi_{ST}$ values were calculated to assess the population structure, while an analysis of the molecular variance (AMOVA) was performed to test the influence of geographical barriers, including the rivers, dry valleys discussed above, and the division between populations of eastern and western cordilleras. A Mantel test of correlation between genetic and geographic distances was performed using 1,000 randomizations (also in Arlequin). Lastly, Structure 2.3.4. (*Pritchard, Stephens & Donnelly, 2000*) was used to estimate the number of populations ($K$), using no admixture for the mtDNA and admixture for the nuclear gene. A total of 10 iterations were run per potential $K$, and the presence of one to eight populations were tested for both beetle lineages.

## Phylogenetic analyses and the molecular clock

Phylogenetic analyses were performed to determine relationships among individuals for each lineage and among lineages. To explore marker-specific effects, mitochondrial and nuclear data sets were initially analyzed separately, but combined data were used to generate the most robust hypotheses for each beetle lineage using MrBayes 3.2 (*Huelsenbeck & Ronquist, 2001*) and RAxML version 8.2.8 (*Stamatakis, 2014*). The analyses performed in MrBayes incorporated a GTR + G + I model, with two independent runs through 1,000,000 generations, with trees sampled every 100th generation. For RAxML (version 8.2.8; *Stamatakis, 2014*), analyses were launched from Mesquite's Zephyr package (*Maddison & Maddison, 2015*) with 1,000 bootstrap replicates using default settings. In addition to these phylogenetic inferences, the number of haplotypes was estimated using DnaSP (*Rozas et al., 2003*) and haplotype networks were inferred with The method of Templeton, Crandall and Sing (TCS) (*Clement, Posada & Crandall, 2000*), using default settings.

To understand the timing and rate of diversification for each ground beetle lineage, divergence times were estimated using an uncorrelated relaxed clock in BEAST 2.0

(*Bouckaert et al., 2014*) using the COI and CAD data matrixes. For *Dyscolus alpinus*, one point of fossil calibration was used to designate minimum node ages, using a log-normal distribution and enforced monophyly. This fossil represents the same tribe as *Dyscolus* (Platynini): a fossil of *Limodromus* from the late Eocene (37.2–33.9 Mya), of Baltic origin (*Schmidt, 2015*). The divergence time for the *Dercylus* lineage was also estimated using two points of fossil calibration: *Amara* (Harpalinae: Zabrini) from the Middle Eocene (44.1 Mya), of Baltic origin (*Larsson, 1978*), and *Pterostichus walcotti* (Harpalinae: Pterostichini) from the Eocene (37.2–33.9 Mya) of Teller County, Colorado (*Scudder, 1900*). In addition to fossil calibrations, substitution rates presented by *Andújar, Serrano & Gámez-Zurita (2012)* for COI were used in both analyses for comparison. Log files from BEAST were examined in Tracer 1.5 (http://tree.bio.ed.ac.uk/software/tracer/) to ensure effective samples sizes. Trees were visualized in TreeAnnotator 2.0.02 (http://beast.bio.ed.ac.uk), using maximum clade credibility after a 10% burn–in.

### Maximum entropy models for species in the genus *Dercylus*

The ecological niche of each species of *Dercylus* was modeled to determine if species distribution overlapped, using present bioclimatic variables at 2.5' spatial resolution (WorldClim v.1.4; *Fick & Hijmans, 2017*) with the use of MAXENT 3.3.3 (*Phillips, Anderson & Schapire, 2006*). *Dercylus* from Cotacachi and Atillo were excluded from these analyses given the few records available. For other species of *Dercylus*, sites mentioned in *Moret (2005)* were georeferenced using GeoLocate (*Rios & Bart, 2010*) and combined with the distribution data generated in this study. A total of 41 occurrence records were gathered for analyses. The model of distribution was based on five bioclimatic variables: $BIO_1$ = Annual Mean Temperature, $BIO_2$ = Mean Diurnal Range (Mean of monthly (max temp–min temp)), $BIO_5$ = Max Temperature of Warmest Month, $BIO_6$ = Min Temperature of Coldest Month, $BIO_{12}$ = Annual Precipitation, $BIO_{13}$ = Precipitation of Wettest Month and $BIO_{14}$ = Precipitation of Driest Month. The analyses ran for 10 replicates of 500 iterations and using a 10% training presence, using the default settings for the convergence threshold ($10^{-5}$). For each species, the output from the average run was reclassified using ArcMap 10.4.1 (Esri, Redlands, CA, USA) with a 10% threshold. Model performance was evaluated using the area under the curve (AUC) calculated by MAXENT, where values between 0.7 and 0.9 indicated a good discrimination (*Swets, 1988*).

## RESULTS

### Genetic diversity and population structure in *Dyscolus alpinus*

The portion of the COI gene analyzed for *Dyscolus alpinus* was 767 bp long, with 41 parsimony informative sites making up six haplotypes (Fig. 2A; GenBank accessions MK440253–MK440258). Haplotype 1 is the most widespread across populations, found in four of six sites (Fig. 2A). Unique haplotypes were recorded from Atillo (H3), Mojanda (H5), and Cayambe (H6; Fig. 2A). Haplotype diversity (*h*) is shown in Table 2 for each population, where Mojanda exhibits the highest haplotypic diversity (*h* = 0.71). The overall nucleotide diversity of COI was low (π = 0.00–0.01). Structure analyses tended to

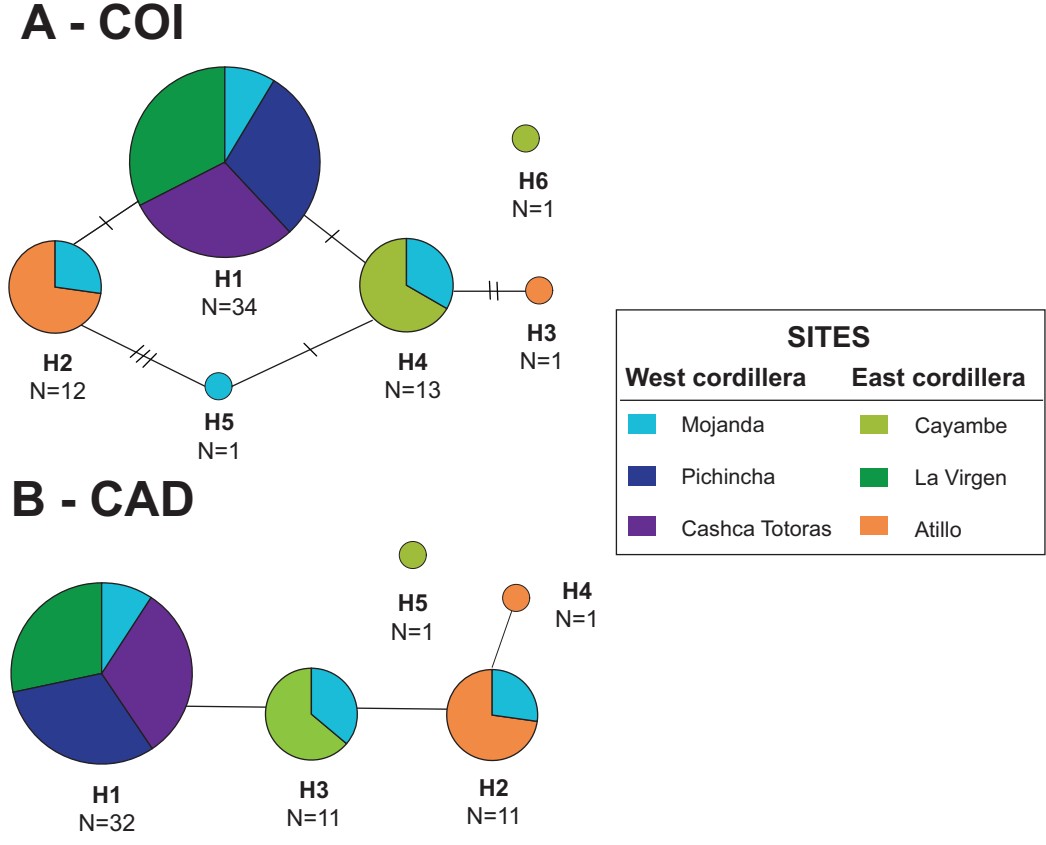

**Figure 2** TCS haplotype network for *Dyscolus alpinus.* (A) COI and (B) CAD.

overestimate the number of populations; results from multiple simulations show that the number of populations using this data set is one ($K = 1$). Results from the neutrality test for the overall values ($D = -0.81$, $p = 0.42$; $F = -2.16$, $p = 0.40$), as well as on most populations are not significant (Table 2). Cayambe and Atillo were exceptions, but these did not exhibit a high proportion of rare alleles, often associated with population expansion (Fig. 2A). Additionally, a Mantel test showed no correlation ($p = 0.13$, $R^2 = 0.08$) between genetic ($\phi_{ST}$) and geographical distances.

The CAD data set for *Dyscolus alpinus* spanned 744 bp, with 64 parsimony informative sites. These represented five haplotypes (GenBank accessions MK440260–MK440264), with no heterozygote individuals observed. Haplotype 1 is found in four of the six populations. Unique haplotypes were recorded from Atillo (H4) and Cayambe (H5). Haplotype networks for both genes show that most haplotypes are present at multiple sites (Fig. 2B). The overall haplotypic ($h$) and nucleotide diversity was low for CAD data set (Table 2). Among all the sites analyzed, Mojanda showed the highest haplotypic diversity ($h = 0.91$). Structure analyses of the nuclear data set showed three populations ($K = 3$). Similar to the results from the mitochondrial data, most neutrality tests using CAD (overall value $D = -0.86$; $p = 0.36$; $F = -5.97$, $p = 0.40$) were not significant, with Cayambe and Pichincha as exceptions. Again, however, these apparent exceptions exhibit

**Table 2 Overview of the genetic diversity indexes for *D. alpinus* and *Dercylus* species and populations.**

| Species | Site | COI | | | | | | CAD | | | | | |
|---|---|---|---|---|---|---|---|---|---|---|---|---|---|
| | | N | h ± SD | π ± SD | S | D | F | N | h ± SD | π ± SD | S | D | F |
| *Dyscolus alpinus* | Mojanda | 11 | 0.71 ± 0.05 | 0.01 ± 0.00 | 10 | 0.87 (p = 0.82) | 1.35 (p = 0.23) | 10 | 0.91 ± 0.04 | 0.01 ± 0.00 | 36 | −1.24 (p = 0.68) | −0.25 (p = 0.18) |
| | Pichincha | 9 | 0.19 ± 0.11 | 0.01 ± 0.00 | 1 | −1.11 (p = 0.17) | −0.34 (p = 0.33) | 11 | 0.40 ± 0.18 | 0.00 ± 0.00 | 2 | −1.40 (p = 0.04)* | −1.16 (p = 0.19) |
| | C. Totoras | 10 | 0.19 ± 0.11 | 0.00 ± 0.00 | 1 | −1.11 (p = 0.19) | −0.34 (p = 0.33) | 10 | 0.20 ± 0.01 | 0.00 ± 0.00 | 1 | −0.59 (p = 0.19) | −0.09 (p = 0.35) |
| | Cayambe | 10 | 0.20 ± 0.15 | 0.01 ± 0.02 | 30 | −2.09 (p = 0.00)** | −5.18 (p = 0.01)** | 8 | 0.23 ± 0.13 | 0.00 ± 0.00 | 11 | −0.85 (p = 0.01)** | −5.96 (p = 0.00)* |
| | La Virgen | 11 | 0.00 ± 0.00 | 0.00 ± 0.00 | 0 | 0.00 (p = 1.00) | 0.00 (p = 1.00) | 11 | 0.00 ± 0.00 | 0.00 ± 0.00 | 0 | 0.00 (p = 1.00) | 0.00 (p = 1.00) |
| | Atillo | 10 | 0.61 ± 0.16 | 0.00 ± 0.01 | 9 | −1.47 (p = 0.05)* | −8.45 (p = 0.00)** | 9 | 0.21 ± 0.12 | 0.00 ± 0.00 | 0 | −1.11 (p = 0.18) | −0.01 (p = 0.36) |
| *Dercylus* | | | | | | | | | | | | | |
| *D. orbiculatus* | Cajas | 10 | 0.51 ± 0.09 | 0.00 ± 0.00 | 2 | 0.33 (p = 0.81) | 0.07 (p = 0.29) | 10 | 0.00 ± 0.00 | 0.00 ± 0.00 | 0 | 0.00 (p = 1.00) | 0.00 (p = 1.00) |
| | Culebrillas | 10 | 0.86 ± 0.07 | 0.01 ± 0.01 | 15 | 2.26 (p = 0.05)* | 4.43 (p = 0.03)* | 9 | 0.21 ± 0.12 | 0.00 ± 0.00 | 1 | −1.08 (p = 0.19) | −0.01 (p = 0.36) |
| *D. cordicollis* | Pichincha | 10 | 0.21 ± 0.12 | 0.00 ± 0.00 | 1 | −1.08 (p = 0.19) | −0.01 (p = 0.36) | 8 | 0.66 ± 0.07 | 0.15 ± 0.00 | 2 | 1.64 (p = 0.95) | 0.96 (p = 0.03)* |
| *D. praepilatus* | Salinas | 8 | 0.00 ± 0.00 | 0.00 ± 0.00 | 0 | 0.00 (p = 1.00) | 0.00 (p = 1.00) | 5 | 0.00 ± 0.00 | 0.0 ± 0.00 | 0 | 0.00 (p = 1.00) | 0.00 (p = 1.00) |
| *Dercylus* sp. | Atillo | 10 | 0.73 ± 0.08 | 0.00 ± 0.00 | 4 | 0.78 (p = 0.66) | 0.86 (p = 0.25) | 10 | 0.35 ± 0.11 | 0.02 ± 0.00 | 1 | 0.01 (p = 0.73) | 0.70 (p = 0.39) |
| *Dercylus* sp. | Cotacachi | 1 | 0.00 ± 0.00 | 0.00 ± 0.00 | 0 | 0.00 (p = 1.00) | 0.00 (p = 1.00) | 1 | 0.00 ± 0.00 | 0.0 ± 0.00 | 0 | 0.00 (p = 1.00) | 0.00 (p = 1.00) |

**Notes:**
*N*, refers to the number of individuals sampled; *S*, number of segregating sites; *h*, haplotypic diversity; *π*, is a measure of nucleotide diversity; *D*, represents the Tajima's *D* a neutrality test statistics with its corresponding *p*-value, and *F*, represents Fu's Fs.
* Indicates significance at *P*-value <0.05.
** Indicates significance at *P*-value <0.01.

few haplotypes, inconsistent with a population expansion scenario. Results from the Mantel test show there was no correlation between genetic and geographic distances ($p = 0.81$, $R^2 = 0.01$).

The AMOVA of nuclear and mitochondrial data for *Dyscolus alpinus* shows that most of the variation occurs within populations (Table 3). These results seem to contrast with the overall $\phi_{ST}$ values (COI $\phi_{ST} = 0.54$; CAD $\phi_{ST} = 0.26$), which suggest moderate levels of population subdivision among populations. When $\phi_{ST}$ values were compared on a population–by–population basis, some genetic differentiation was found among populations (Table 4; Fig. 3). COI data exhibited higher overall genetic differentiation between sites ($\phi_{ST} = 0.32$–$0.90$; Table 4). Some exceptions are found between sites that are closer in distance (less than 36 km), such as Cayambe–Mojanda (separated by 30 km): $\phi_{ST} = 0.10$, and La Virgen-Pichincha (separated by 36 km): $\phi_{ST} = −0.10$. The CAD data set shows lower genetic differentiation among all populations of *Dyscolus alpinus* (Table 4). However, higher $\phi_{ST}$ values are observed when Cashca Totoras is compared with

Table 3 Analysis of the molecular variance for populations of *D. alpinus*.

| No. groups | Partitions | Tests | COI | | | CAD | | |
|---|---|---|---|---|---|---|---|---|
| | | | Among groups | Among populations | Within populations | Among groups | Among populations | Within populations |
| 2 | (1, 2, 4) (6, 7, 8) | E–W | −1.51 | 3.96 | 97.55 | −1.51 | 3.96 | 97.55 |
| 2 | (1, 2, 6, 7) (8, 7) | N–S | −0.41 | 19.64 | 80.77 | 2.47 | 1.65 | 95.88 |
| 2 | (8) (1, 2, 4, 6, 7) | 7 vs all | 11.5 | 14.32 | 74.18 | −5.91 | 5.21 | 100.7 |
| 2 | (4) (1, 2, 6, 7, 8) | 4 vs all | −8.14 | 23.15 | 84.99 | 24.89 | −6.11 | 81.22 |
| 4 | (1, 2) (4) (6, 7) (8) | Barriers 1 | −10.11 | 28.36 | 81.76 | 10.31 | −6 | 95.69 |
| 3 | (1, 2)(4) (6, 7, 8) | Barriers 2 | −13.96 | 30.42 | 83.54 | 12.39 | −6.31 | 93.92 |
| 4 | (1, 2, 4) (6) (7) (8) | Barriers 3 | 10.91 | 10.36 | 78.74 | −11.65 | 12.18 | 99.47 |

Table 4 $\phi_{ST}$ values of *D. alpinus* for COI (lower diagonal) and CAD (upper diagonal) genes.

| Populations | 1 | 2 | 4 | 6 | 7 | 8 |
|---|---|---|---|---|---|---|
| 1. MO | – | −0.02 | 0.50 | 0.22 | 0.22 | 0.15 |
| 2. PI | 0.50 | – | 0.53 | 0.31 | 0.01 | 0.00 |
| 4. CT | 0.32 | 0.90 | – | −0.08 | 0.65 | 0.62 |
| 6. CY | 0.10 | 0.56 | 0.40 | – | 0.40 | 0.40 |
| 7. LV | 0.50 | −0.10 | 0.88 | 0.55 | – | 0.01 |
| 8. AT | 0.40 | 0.86 | 0.81 | 0.50 | 0.85 | – |

the other sites ($\phi_{ST} = 0.5–0.65$; Fig. 3), except when compared to Cayambe ($\phi_{ST} = –0.08$). The overall number of migrants between *Dyscolus* populations (Nm = 0.31) indicates that there is population subdivision across sites. Higher levels of genetic connectivity were observed between Mojanda and Cayambe (Nm = 1.79), results that are also supported by the low $\phi_{ST}$ values between these two sites (Table 4).

## Genetic diversity and population structure in species of the genus *Dercylus*

In the case of *Dercylus* species, the portion of the COI gene analyzed was 776 bp, with 76 parsimony informative sites constituting 16 haplotypes (GenBank accessions MK440220–MK440235). No haplotypes were shared among sites (Fig. 4A), and overall nucleotide diversity within these species is low ($\pi = 0.00–0.01$; Table 2). Eight haplotypes were recorded for populations of *Dercylus orbiculatus*, whereas two haplotypes were recovered from *Dercylus cordicollis*. The individual sampled from Cotacachi represents a distinct lineage of *Dercylus* closely related to *Dercylus cordicollis*, based on the TCS analyses and phylogenetic inference, though only one haplotype was found. For specimens of *Dercylus praepilatus* from Salinas, one haplotype was recorded, while four haplotypes were recorded from specimens collected from the locality of Atillo. Based on the morphological similarity with *Dercylus orbiculatus*, specimens from Atillo were thought to be another population of *Dercylus orbiculatus*; however, based on haplotype designation and other phylogenetic analyses, this population appears to be a distinct lineage/species of *Dercylus*.

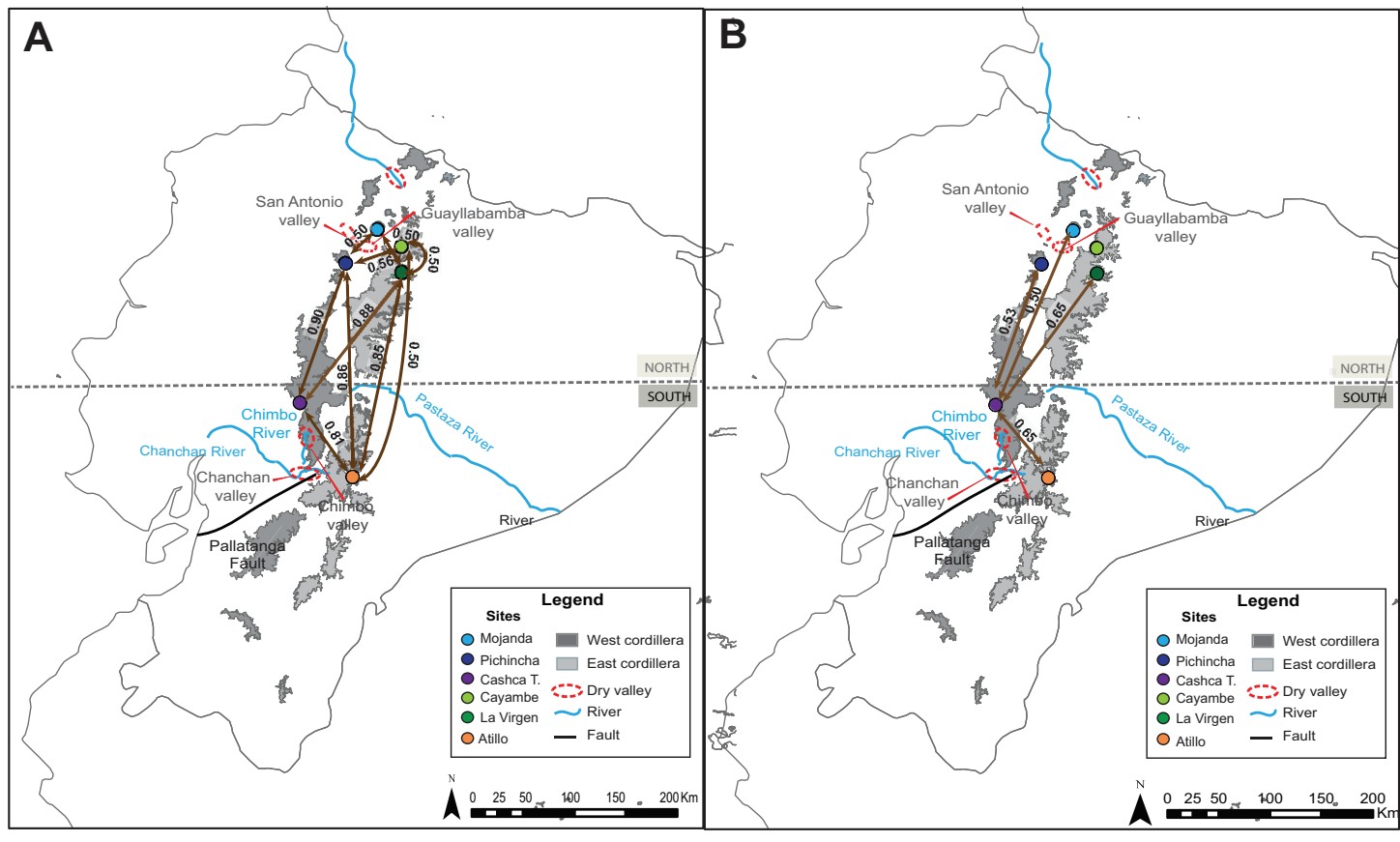

**Figure 3** $\phi_{ST}$ **values among populations of *Dyscolus alpinus*.** (A) COI and (B) CAD. Full-size $\square$ DOI: 10.7717/peerj.7226/fig-3

Haplotype diversity indices are shown in Table 2 for each species and population, where populations of *Dercylus orbiculatus* from Culebrillas show the highest haplotypic diversity ($h = 0.86$) among *Dercylus* species and populations. Structure analyses of *Dercylus orbiculatus* individuals all resolved as one population, even though the two sites do not share haplotypes. These analyses were also performed for the entire clade, and one population was found ($K = 1$). Neutrality test values were not significant (overall value $D = 0.26$, $p = 0.77$; $F = 0.89$, $p = 0.44$), with the exception of the population of *Dercylus orbiculatus* from Culebrillas, for which significantly positive $D$ and $Fs$ were found (Table 2). Lastly, the Mantel test performed with $\phi_{ST}$ and geographic distances show no significant correlation ($R^2 = 0.05$, $p = 0.3$) between genetic and geographical distances.

The portion of the CAD gene amplified for *Dercylus* species was 669 bp with 24 parsimony informative sites (GenBank accessions MK440236–MK440251). A total of 13 alleles were identified across the *Dercylus* lineage and the overall nucleotide diversity was low ($\pi = 0.00$–$0.02$). Four alleles were recovered from *Dercylus orbiculatus* populations, and three alleles from *Dercylus cordicollis*, with one shared with *Dercylus* from Cotacachi (H5; Fig. 4B). Four alleles were identified in the population of *Dercylus* from Atillo, and three from *Dercylus praepilatus*. Two alleles were shared between *Dercylus praepilatus* and *Dercylus* sp. from Atillo (H8 and H11; Fig. 4B). The highest allele diversity is recorded

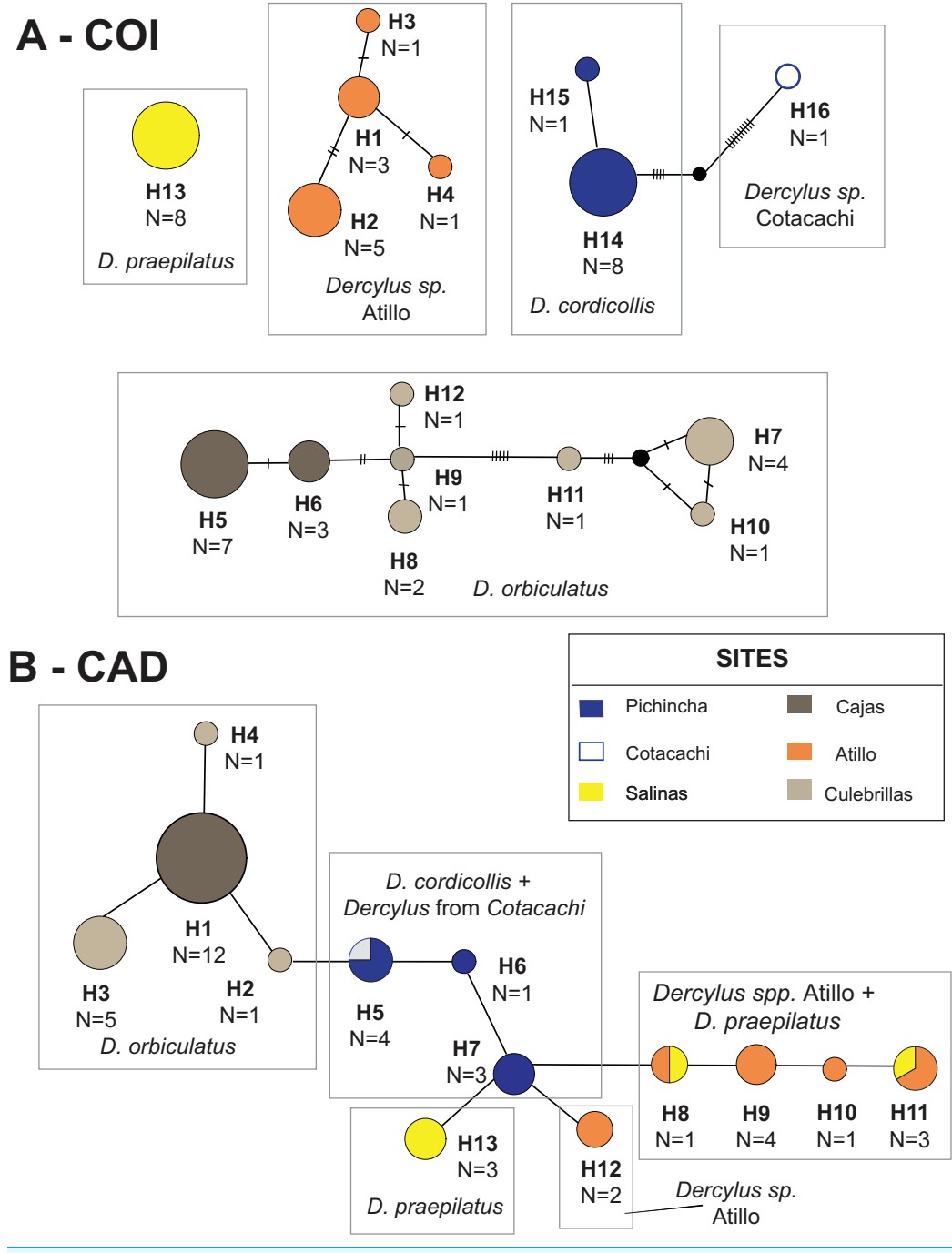

**Figure 4 TCS haplotype network for species in *Dercylus* lineage.** (A) COI and (B) CAD.

for the *Dercylus* sp. from Atillo, followed by *Dercylus orbiculatus* from El Cajas and *Dercylus cordicollis* (Table 2). Structure analyses for *Dercylus orbiculatus* populations alone showed only one population ($K = 1$), whereas results from the analyses of the whole clade show five distinct populations ($K = 5$), corresponding to each species. Neutrality tests supported significantly positive *Fs* for *Dercylus cordicollis* from Pichincha (Table 2).

**Table 5 Analysis of the molecular variance for populations and species of *Dercylus*.**

| No. groups | Partitions | Tests | COI | | | CAD | | |
|---|---|---|---|---|---|---|---|---|
| | | | Among groups | Among populations | Within populations | Among groups | Among populations | Within populations |
| 2 | (2, 3, 5, 10) (8, 9) | E–W | −11.28 | 102.46 | 8.82 | −63.40 | 154.07 | 9.34 |
| 2 | (2, 3, 10) (5, 8, 9) | N–S | 11.89 | 80.09 | 8.02 | 30.61 | 62.93 | 6.46 |
| 2 | (8) (2, 3, 5, 9, 10) | 8 vs all | 35.57 | 55.23 | 9.20 | −6.07 | 98.29 | 7.76 |
| 4 | (2, 10) (3) (5, 9) (8) | Barriers 1 | 66.33 | 25.88 | 7.78 | 93.82 | −0.37 | 6.55 |
| 3 | (2, 10, 3) (5) (9) (8) | Barriers 2 | 10.86 | 80.79 | 8.35 | 94.46 | −0.60 | 6.14 |
| 5 | (2, 10) (3) (5) (9) (8) | Barrier 3 | 22.59 | 68.99 | 8.42 | 96.81 | −4.26 | 7.45 |
| | (2) (10) (3) (5,9) (8) | Barriers 4 | 73.18 | 19.05 | 7.78 | 93.12 | 0.25 | 6.63 |
| 2 | (5, 9) (3, 5, 8, 10) | 8,9 vs all | 26.22 | 10.43 | 11 | −31.50 | 122.97 | 8.53 |

**Table 6 $\phi_{ST}$ values for species of *Dercylus* COI (lower diagonal) and CAD (upper diagonal) genes.**

| Populations | 2 | 3 | 5 | 8 | 9 | 10 |
|---|---|---|---|---|---|---|
| 2. *D. cordicollis*—Pichincha | – | 0.00 | 1.00 | 0.06 | 0.95 | 0.00 |
| 3. *D. praepilatus*—Salinas | 0.99 | – | 1.00 | 0.03 | 0.95 | 0.00 |
| 5. *D. orbiculatus*—Cajas | 0.98 | 0.60 | – | 0.93 | 0.10 | 1.00 |
| 8. *Dercylus* spp. Atillo | 0.96 | 0.97 | 0.96 | – | 0.90 | −0.77 |
| 9. *D. orbiculatus*—Culebrillas | 0.80 | 0.84 | 0.60 | 0.85 | – | 0.92 |
| 10. *Dercylus* spp. Cotacachi | 0.98 | 1.00 | 0.98 | 0.95 | 0.71 | – |

Results from the Mantel test show no significant correlation ($p = 0.8$, $R^2 = 0.02$) between genetic ($\phi_{ST}$) and geographical distances.

The AMOVA showed that most of the genetic variation was found among groups when testing the effect of geographical barriers (Table 5). The high variation among groups is consistent with the existence of multiple species, since species in this genus do not have widespread distributions, with the exception of *Dercylus orbiculatus*, occurring across the southern Ecuadorian Andes (Fig. 1; *Moret, 2005*). The highest percentage of variation in the AMOVA when parsing the data by species was seen across the Chimbo and Chanchan dry valleys and rivers, which overlap with another potential geographical barrier, the Pallatanga depression. These results were also supported by the overall $\phi_{ST}$ values, which show high levels of differentiation among species and sites (COI $\phi_{ST} = 0.89$; CAD $\phi_{ST} = 0.95$). A high level of genetic differentiation was also observed when $\phi_{ST}$ values were compared between populations and species in COI data set (Table 6). In this data set, the lowest $\phi_{ST}$ values were observed between populations of *Dercylus orbiculatus* from Cajas and Culebrillas ($\phi_{ST} = 0.50$).

In the case of the nuclear coding gene CAD, the overall value of $\phi_{ST}$ ($\phi_{ST} = 0.93$) shows high levels of population differentiation. When each population was compared against each other, we found moderate to complete population subdivision between some sites ($\phi_{ST} = 0.77–1.00$; Table 6), particularly when comparing against populations of *Dercylus*

*orbiculatus* from Cajas or Culebrillas. Lower $\phi_{ST}$ values were found between the two populations of *Dercylus orbiculatus* Cajas and Culebrillas ($\phi_{ST} = 0.10$), as well as between Pichincha and Cotacachi ($\phi_{ST} = 0.00$), Pichincha and Salinas ($\phi_{ST} = 0.00$), Salinas and Atillo ($\phi_{ST} = 0.03$), and Salinas and Cotacachi ($\phi_{ST} = 0.00$). The number of migrants per generation was calculated for two populations of *Dercylus orbiculatus* (Cajas and Culebrillas), which revealed low gene flow (Nm = 0.39).

## Phylogenetic analyses and divergence time estimations

Phylogenetic analyses were conducted separately for each gene data set using Bayesian and Maximum Likelihood methods. Analyses of the two fragments for *Dyscolus alpinus* yields similar resolution (Figs. 5A and 5B), with *Dyscolus* sp. (SIMT208) as sister to the *Dyscolus alpinus* clade (Fig. 5A). These results are consistent with the TCS haplotype network, which shows most samples are contained within one network, with the exception of COI haplotype 6 (SIMT208–H6; Fig. 2A). Most haplotypes are present at multiple sites (Fig. 5A), with only two unique haplotypes (Atillo-H3 and Mojanda-H5). Haplotype 1 is found at five of six sampling sites (Mojanda, Cayambe, Pichincha, La Virgen, Cashca Totoras); haplotype 2 and haplotype 4 are each found at two (Atillo, Mojanda and Cayambe, Mojanda, respectively; Fig. 2A). The highest haplotypic diversity was found at Mojanda with four haplotypes (H1, H2, H4, H5), while the lowest was at Cashca Totoras and Pichincha with only the widespread H1 haplotype found at each site. A similar pattern was recovered for CAD. First, CAD also suggested that SIMT208 from Cayambe represented a distinct *Dyscolus* species (Fig. 5B). Within *Dyscolus alpinus*, four CAD haplotypes were recorded (Fig. 5B). Most sites yielded haplotype 1 (Mojanda, Cayambe, Pichincha, La Virgen, Cashca Totoras, Fig. 2B), with only one unique haplotype (H4) observed from an individual from Atillo. The Mojanda population showed the highest haplotypic diversity (H1, H2, H3), whereas Pichincha (H1) and Cashca Totoras (H1) each only revealed one widespread haplotype. A relaxed molecular clock was used on the combined matrix for *Dyscolus alpinus* to calculate divergence time estimates for the clade (Fig. 6). These analyses suggest that the *Dyscolus alpinus* clade originated during the Miocene 6.09 Mya (1.26–13.64 Mya). The timing of splitting–events for the *Dyscolus alpinus* clade (14.33 Mya, with a confidence interval of 3.63–27.46 Mya; Fig. 6) is slightly older than most páramo plant species (2–5 Mya; *Madriñán, Cortés & Richardson, 2013*), but contemporary with the evolution of highland species in the northern Andes mountain chain, which arose during the Miocene (*Weir, 2006*; *Hines, 2008*).

For *Dercylus*, the two genes reveal slightly different patterns (Figs. 7A and 7B). The COI tree suggests that most species and sites are monophyletic (Fig. 7A), with *Dercylus orbiculatus* the exception, appearing as a paraphyletic grade. The CAD gene tree shows less structure associated with site and species, since multiple alleles are shared among species, for example, between *Dercylus praepilatus* and *Dercylus* sp. from Atillo (H8, H11), and between *Dercylus cordicollis* and *Dercylus* sp. from Cotacachi (H5; Fig. 4B). Yet, in CAD, *Dercylus orbiculatus* appears as a distinct clade, supported by bootstrap and posterior probability values (Fig. 7B).

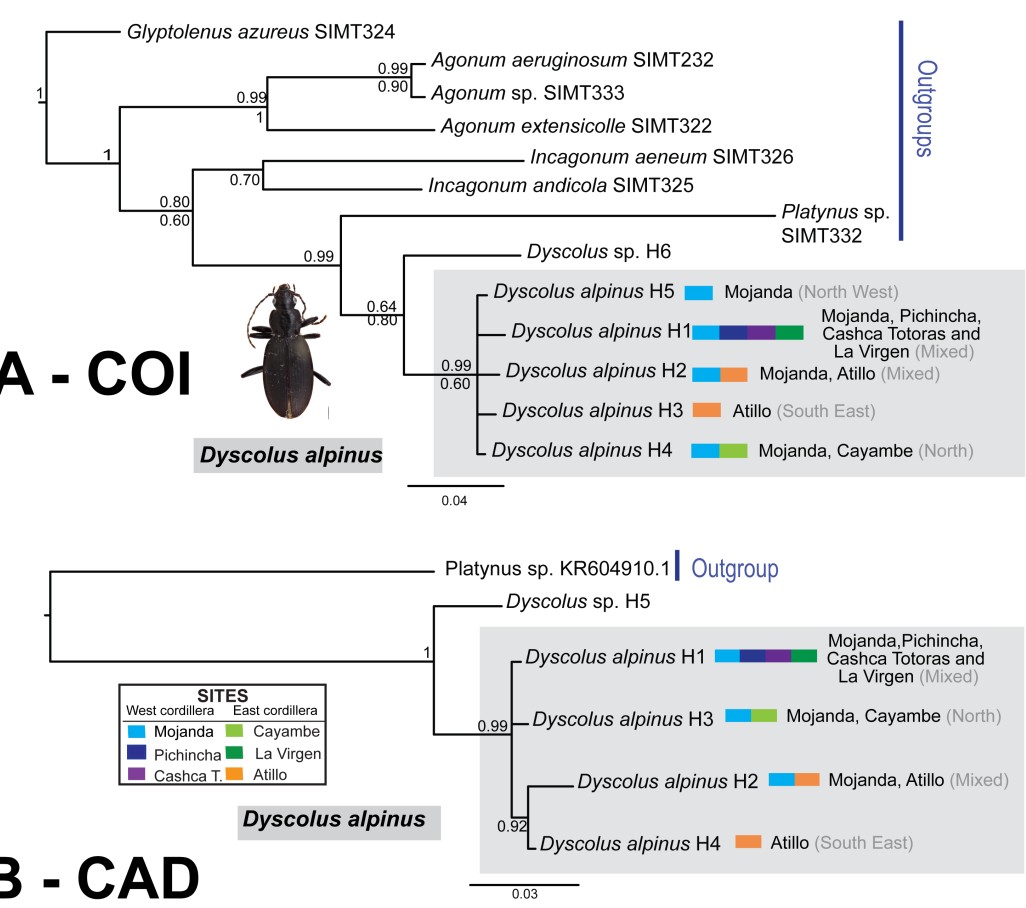

**Figure 5 Bayesian 50% rule consensus tree for *D. alpinus*.** (A) COI and (B) CAD, were phylogenetic analysis included one sample of each haplotype. Site information for each haplotype is presented as colored bars for each gene tree, and in parentheses directionally (North, North West, and South East) is shown, and haplotypes present in multiple sites were labeled as mixed. Posterior probabilities are shown above the branches and bootstrap support values for the ML tree are shown below branches.

The analysis of the combined matrix (COI and CAD; Fig. 8) exhibits clearer resolution among species and populations of *Dercylus*, though *Dercylus orbiculatus* appears again as a paraphyletic grade, as in the COI tree. A relaxed molecular clock based on the combined data set shows that the *Dercylus* clade originated during the Oligocene 29.81 Mya, with a confidence interval of 12.45–50.38 Mya. Thus, most of the sampled species in this study originated prior to the evolution of páramo, with *Dercylus orbiculatus* (10.74–13.5 Mya, confidence interval of 1.10–28.58 Mya), *Dercylus* from Cotacachi (16.39 Mya, confidence interval of 3.60–24.38 Mya) and *Dercylus* sp. from Atillo (6.04 Mya, originating confidence interval of 1.13–12.90 Mya) during the Miocene (16.4–6.04 Mya). *Dercylus cordicollis*, on the other hand, originated in the Pliocene (3.68 Mya, confidence interval of 0.00–6.96 Mya), and *Dercylus praepilatus* (2.38 Mya, confidence interval of 0.01–7.23 Mya) in the Pleistocene. Only these last two diversifications were contemporary with the diversification of most páramo plants (2–5 Mya, *Madriñán, Cortés & Richardson, 2013*).

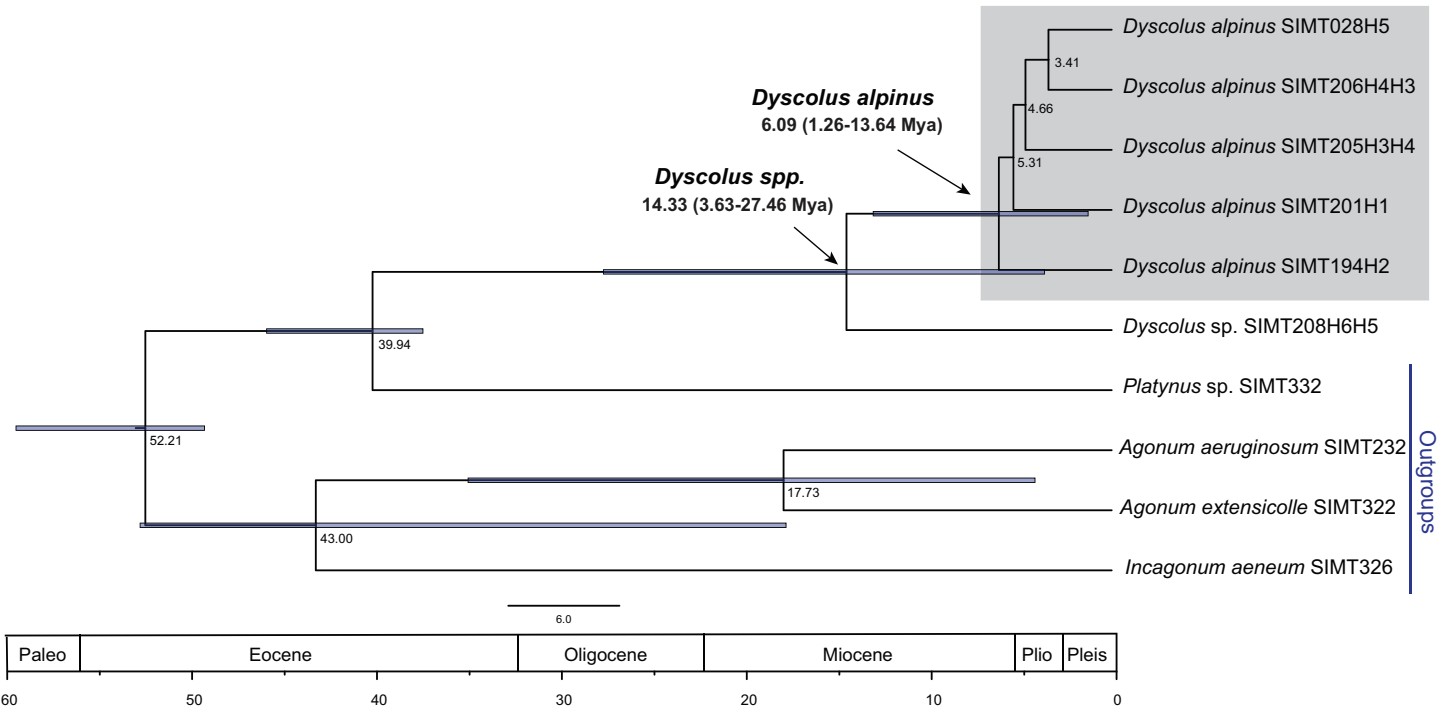

**Figure 6 Timing of the *D. alpinus* clade lineage based on a relaxed molecular clock for both genes.** Voucher numbers (Table S1) and haplotype designation are recorded at the end for each member of *Dyscolus* clade. For each sample the first haplotype corresponds to the mitochondrial data set, followed by the nuclear haplotype (e.g., SIMT208H6H5), with exception of SIMT028 were only the mtDNA was amplified.

## Niche modeling for species in the genus *Dercylus*

The ecological niche model for the three described species of *Dercylus* showed a high predictive ability under a random model, with AUC values between 0.998–0.999. Populations of this ground beetle lineage are mainly present in the Andean region of Ecuador, with potentially overlapping distributions (Fig. 9). Compared to other species analyzed, the maximum entropy model for *Dercylus orbiculatus* shows that this species is present at high elevation, but its distribution also extends to lower elevations, especially toward the south. In contrast, *Dercylus praepilatus* has the smallest potential distributional range. Lastly, niche modeling also revealed that suitable habitats for species of this genus also extend into to the Andean regions of Colombia and Peru. To test if the distribution of the species extends into these areas, further sampling is needed.

## DISCUSSION

The effect of dispersal ability on population genetic structure and diversification represents a rich focus of study (*Bohonak, 1999*). Insects, varying so broadly in body size and flight ability, have played a prominent role in such studies (*Brühl, 1997*; *Gutiérrez & Menéndez, 1997*; *Ikeda, Nishikawa & Sota, 2012*). Yet, apparently limited dispersal ability (through wing reduction/loss) has been associated with varied results, from high phylogeographic structure (*Gutiérrez & Menéndez, 1997*; *Ikeda, Nishikawa & Sota, 2012*), to low levels of gene flow and only moderate structure (*Chatzimanolis & Caterino, 2007*;

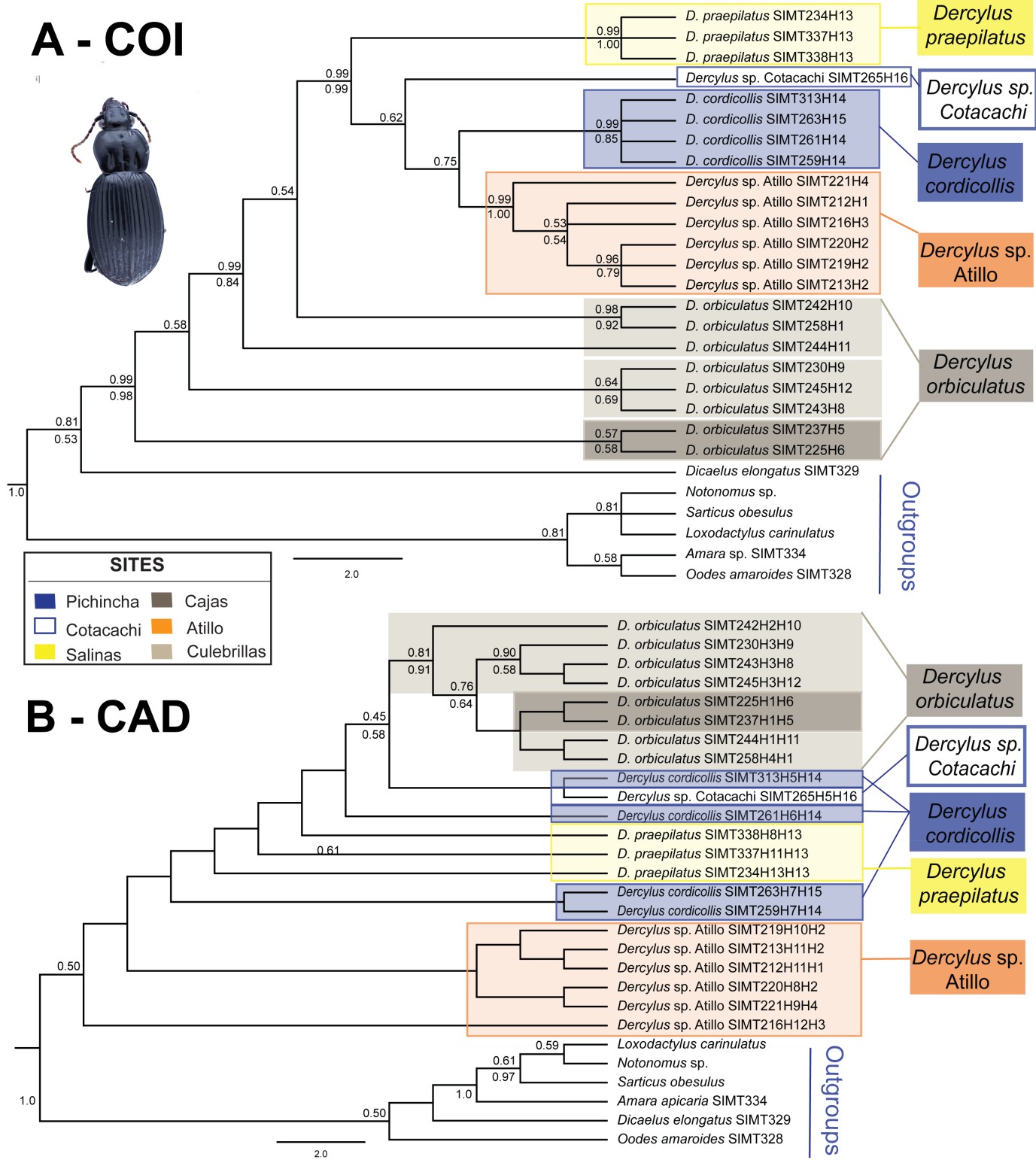

**Figure 7 Bayesian 50% rule consensus tree for species of *Dercylus*.** (A) COI and (B) CAD. Posterior probabilities are shown above the branches and bootstrap support values for the ML tree are shown below branches. Colored boxes around clades indicate localities as shown in the legend.

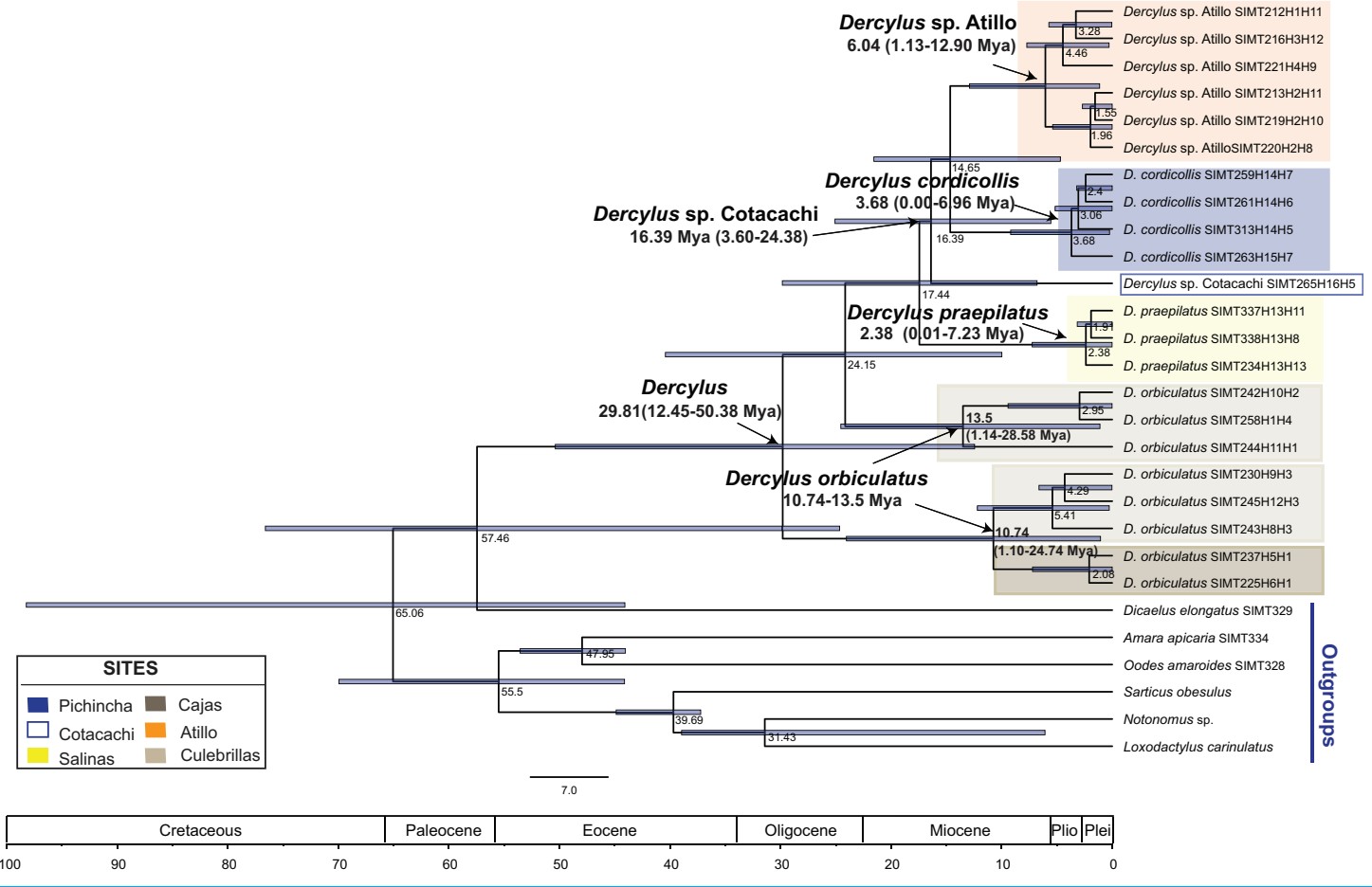

**Figure 8 BEAST tree for *Dercylus* based on a relaxed molecular clock for both genes.** Voucher number (Table S1) and haplotype designation are recorded at the end of each individual in the genus *Dercylus*. For each sample the first haplotype corresponds to the mitochondrial data set, followed by the nuclear haplotype (e.g., SIMT212H1H11).

*Huang & Lin, 2010*). In tropical mountains, wingless and wing dimorphic species dominate beetle communities (*Erwin, 1985*; *Ashe & Leschen, 1995*; *Moret, 2005*), but the relationship between wing reduction and diversification in the tropical Andes has been little explored. The present study is among a very few (*Hines, 2008*; *Maddison, 2014*; *De-Silva et al., 2016*) focused on insects from the Andean region, an area characterized by its high diversity (*Myers et al., 2000*; *Veblen, Young & Orme, 2015*). Our analyses of two flightless ground beetles from páramo show different degrees of population subdivision. *Dyscolus alpinus*, which exhibits some wing polymorphism (from micropterous to brachypterous individuals), shows a low haplotypic diversity across sampled populations. In *Dercylus* (entirely micropterous), however, high levels of differentiation were found at the species and population levels. The reasons for these differences are not entirely clear.

In *Dyscolus alpinus*, a dominant haplotype for each marker is found in four out of the six populations analyzed, with one site (Mojanda) exhibiting unusually high diversity. The diversity at this site could be associated with multiple colonization events due to dynamic habitat shifts from volcanic activity dated prior to the Last Glacial Maximum

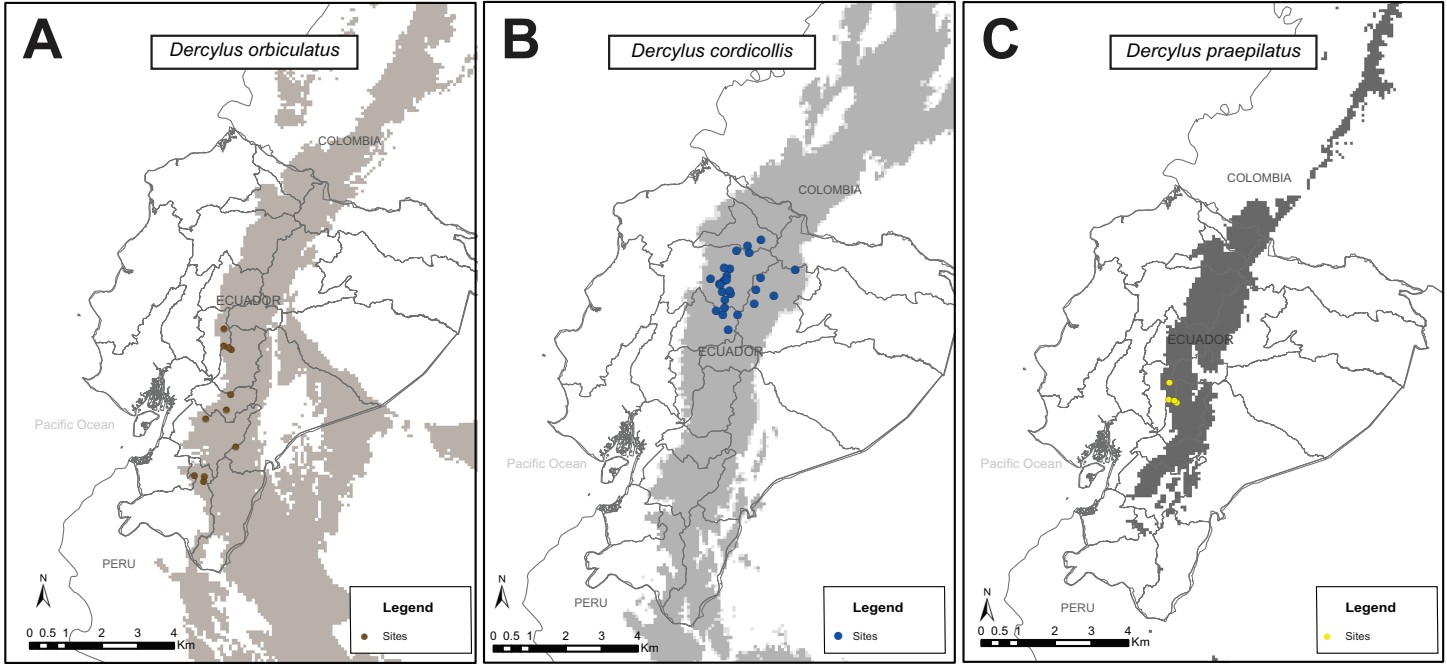

**Figure 9 Maximum entropy models for present distribution for species in the genus *Dercylus* across the Ecuadorian Andes using five bioclimatic variables. (** A) *D. orbiculatus*, (B) *D. cordicollis* and (C) *D. praepilatus*, Colored points represent localities included in the model. Shaded areas represent the broader predicted distribution across the northern Andes.

(Mt. Mojanda and Mt. Fuya Fuya, *Robin et al., 2009*), and periodic burns during the Holocene (*Frederick et al., 2018*). Even though most genetic variation occurs within populations of *Dyscolus alpinus*, some among–population genetic differentiation was also revealed, particularly comparing Cashca Totoras (site 4) to other sites. Some population pairs on opposite sides of the cordillera also present high population structure (e.g., Pichincha–Atillo), but this is not the case for most sites, so simple E–W and N–S groupings show minimal explanatory power. High $\phi_{ST}$ values found among sites on the same side of the cordillera could have resulted from habitat discontinuity reinforced by the presence of major geographical barriers such as major rivers and dry valleys (*Krabbe, 2008*; *Guayasamin et al., 2010*; *Quintana et al., 2017*). For example, in the eastern cordillera the Atillo vs Cayambe and La Virgen populations are separated by the Pastaza river, previously proposed as a limiting factor for some species in the Ecuadorian Andes (*Krabbe, 2008*; *Guayasamin et al., 2010*). In addition to major rivers, the presence of lower elevation dry valleys appears to be contributing to higher structure between northern populations. This is the case for the Pichincha and Mojanda populations, located relatively close in distance (~47 km), but separated by the Guayllabamba valley (*Quintana, 2010*). So, rapid changes in elevation and abiotic conditions over short distances do appear to affect gene flow between sites. The dissimilarity in the results from nuclear and mitochondrial data sets in the population genetic analyses is likely associated with differences in the rate of nucleotide change between these markers (*Lin & Danforth, 2004*). The low levels of genetic variation in the CAD data set reflects the recent origin of the *Dyscolus alpinus*

(6.32 Mya, Miocene), whereas the higher population structure in the COI data set probably results from both higher mutation rates and shorter coalescence times of mitochondrial DNA (*Lin & Danforth, 2004*).

For the genus *Dercylus* represented here by multiple (described and undescribed) species, high levels of differentiation are reported at the species and population levels for both genes. The mitochondrial marker showed no shared haplotypes among sites and species, whereas CAD showed three shared haplotypes, generally shared by proximate populations (Salinas + Atillo, and Pichincha + Cotacachi). Species in the genus *Dercylus* show smaller distributions (*Moret, 2005*). However, results from niche modeling for three species of *Dercylus* using present conditions show that these species could have potentially overlapping distributions. Both dispersal ability and geographical barriers appear to be playing roles shaping the genetic diversities of *Dercylus* species. $\phi_{ST}$ values were high even between populations of *Dercylus orbiculatus* (Cajas and Culebrillas, divided E–W by the principal inter–Andean valley), sharing no haplotypes in either marker. It is, however, intriguing that the three CAD haplotypes at Culebrillas appear to be derived from the single common one at Cajas. This may be the basis for Structure only recovering one population ($K = 1$) with COI and CAD sequences, suggesting that in the past these populations had higher connectivity levels. Based on AMOVA results, the Chimbo and Chanchan river valleys, which overlap with the Pallatanga fault (*Baize et al., 2015*), appear to be major geographical barriers for the members of *Dercylus* in the southern Ecuadorian Andes, separating distantly related species. The high mountains between Atillo and Culebrillas (Mt. Ayapungo and Mt. Coyay, above 4,600 m) also seem to have contributed toward the genetic heterogeneity seen in *Dercylus* in the eastern cordillera, considering the high $\phi_{ST}$ values (Table 6) and the lack of shared haplotypes between these two sites (Fig. 4). Lastly, the combination of distance between sites and the presence of lower elevation dry valleys appear to have had a combined effect, separating *Dercylus cordicollis* (Pichincha) and *Dercylus* sp. (from Cotacachi). Divergence time estimates show that the species in the *Dercylus* clade originated during the Miocene, Pliocene and Pleistocene (16–2 Mya; Fig. 8), mostly during the Miocene prior to appearance of páramo (2–5 Mya Pliocene and Plesitocene: *Madriñán, Cortés & Richardson, 2013*), with exception of *Dercylus cordicollis* (3.68 Mya, Pliocene) and *Dercylus praepilatus* (2.38 Mya, Pleistocene).

While phylogenetic and population analyses of *Dercylus* largely support species distinctions, the status of *Dercylus orbiculatus* (sampled from both Cajas and Culebrillas) is less clear. It is resolved as a paraphyletic grade by COI, as well as in the combined data tree, but monophyletic by CAD, with strong support. While this difference may be a result of the difference in rate of nucleotide change between genes (*Lin & Danforth, 2004*), divergences in mtDNA sequences are low (0.25%), network analyses offer no indication that these should be considered distinct.

When results from these two flightless lineages were compared to the patterns observed in *Pelmatellus columbianus*, a macropterous ground beetle also present in páramo (*Muñoz-Tobar, in press*), a common pattern for the distribution of the genetic diversity for ground beetles that live in páramo does not emerge. The flightless beetles analyzed

here both have higher levels of population structure when compared to *Pelmatellus columbianus*. Yet, even though both *Dyscolus* and *Dercylus* lineages are flightless, *Dyscolus alpinus* appears to be a better disperser than species in the genus *Dercylus*. Evidence for this was found in the levels of genetic connectivity between northern populations of *Dyscolus alpinus* that show some levels of gene flow (e.g., Mojanda–Cayambe), perhaps facilitated by Quaternary glaciations, when páramo moved to lower elevation areas such as the inter–Andean valleys (*Villota & Behling, 2014*). Still, the low genetic differentiation for some populations of *Dyscolus alpinus* could also be explained by the relatively recent origin of the lineage (6.09 Mya, comparable to some of the individual species within *Dercylus*). How, then, *Dyscolus alpinus* has achieved its broad distribution, while *Dercylus* species have largely remained confined to restricted areas, remains an open question.

Understanding the history of movements and diversification of organisms in alpine tropical ecosystems has considerable importance, given the effects that climate warming is having on high elevation faunas (*Moret et al., 2016*). Beetle populations in alpine regions of the Andes, including two of the species in this study, are experiencing shifts in elevational range (100–400 m upslope; from grass páramo to super páramo; *Moret et al., 2016*). A clear picture of the diversity, distributions and natural history of these páramo species could have important implications toward the conservation of alpine faunas.

## CONCLUSIONS

The effects of mountain isolation vary among páramo species. The genetic diversity found in both ground beetle lineages analyzed to date appears to be influenced by the range size and dispersal capability for each beetle lineage. While macropterous species such as *Pelmatellus columbianus* exhibit broad elevational ranges (from the inter–Andean valleys to páramo, 2,000– 4,200 m) and sustain higher levels of gene flow among populations (*Muñoz-Tobar, in press*), flightless ground beetles display smaller elevational ranges (2,750–4,200 m) and different degrees of genetic connectivity among populations, from higher levels of genetic connectedness in populations of *Dyscolus alpinus*, to high levels of differentiation among species and populations within the *Dercylus* lineage. Although species in the genus *Dercylus* show similar predicted distributional ranges, less genetic cohesiveness was found among populations, possibly as a result of division of the ecosystem by abrupt topographic features. Species with smaller elevation ranges are probably more susceptible to climate change, and elevational shifts toward higher elevations have been already been reported among ground beetle species from páramo (*Moret et al., 2016*). The patterns of diversification found from the study of páramo ground beetles have important implications for the conservation of species in this ecosystem, allowing conservation efforts to take into consideration local and regional patterns of diversity. These results show that the loss of wings in flightless ground beetles has not completely limited the dispersal of individuals across páramo patches. However, this study only included two lineages within the ground beetles, analyses of other dispersal-limited organisms is necessary to test these inferences.

## ACKNOWLEDGEMENTS

We are very grateful to the people and institutions who assisted us during the field portion of this study or provided samples for analyses: Andrés Romero–Carvajal (Pontificia Universidad Católica del Ecuador), Shelley Langton–Myers (Clemson University), Pierre Moret (Université Toulouse), Anthony Deczynski (Clemson University), Catalina Bravo (INIAP), Max Ochoa (INIAP) and Rosario Tobar–Ocaña (INIAP). Finally, we thank three anonymous reviewers for feedback that significantly improved the manuscript.

### Funding

This work was supported by the King Research Grant (Clemson University) and the Ecuadorian Secretary of Higher Education, Science, Technology and Innovation (SENESCYT). The funders had no role in study design, data collection and analysis, decision to publish, or preparation of the manuscript.

### Grant Disclosures

The following grant information was disclosed by the authors:
King Research Grant: Clemson University.
Ecuadorian Secretary of Higher Education, Science, Technology and Innovation: SENESCY.

### Competing Interests

The authors declare they have no competing interests.

### Author Contributions

- Sofia I. Muñoz-Tobar conceived and designed the experiments, performed the experiments, analyzed the data, contributed reagents/materials/analysis tools, prepared figures and/or tables, authored or reviewed drafts of the paper.
- Michael S. Caterino conceived and designed the experiments, analyzed the data, contributed reagents/materials/analysis tools, authored or reviewed drafts of the paper, approved the final draft.

### Field Study Permissions

The following information was supplied relating to field study approvals (i.e., approving body and any reference numbers):

Research and export permits were granted by the Ministry of the Environment Ecuador (permit number MAE–DNG–ARGG–CM–2014–004), with support of the Pontifical Catholic University of Ecuador and Clemson University.

### DNA Deposition

The following information was supplied regarding the deposition of DNA sequences:

COI sequences described in this manuscript are accessible via GenBank accession numbers MK440220–MK440235 for species in the genus Dercylus, and

MK440253–MK440258 for *Dyscolus alpinus*. In the case of CAD gene, GenBank accession numbers MK440236–MK440251 correspond to species in the genus Dercylus, and MK440260–MK440264 to *Dyscolus alpinus*. Lastly, Genbank accession numbers for the outgroup taxa are MK440259, MK457690–MK457693 for COI gene, and KR604910, MK457694–MK457697 for the CAD gene.

Sequence information is also available in Table S1.

## Data Availability

Muñoz-Tobar, Sofia; Caterino, Michael S. (2019): *Dercylus praepilatus* data points. figshare. Dataset. DOI 10.6084/m9.figshare.8139539.v2.

Muñoz-Tobar, Sofia; Caterino, Michael S. (2019): *Dercylus orbiculatus* data points. figshare. Dataset. DOI 10.6084/m9.figshare.8139536.v2.

Muñoz-Tobar, Sofia; Caterino, Michael S. (2019): *Dercylus cordicollis* data points. figshare. Dataset. DOI 10.6084/m9.figshare.8135414.v1.

## Supplemental Information

Supplemental information for this article can be found online at http://dx.doi.org/10.7717/peerj.7226#supplemental-information.

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
