# Peer review of "The role of dispersal for shaping phylogeographical structure of flightless beetles from the Andes"

_PeerJ, doi:10.7717/peerj.7226_

## Round 0.1 · original submission · Major Revisions

· Academic Editor

Major Revisions

Dear Drs. Muñoz-Tobar and Caterino:

Thanks for submitting your manuscript to PeerJ. I have now received three independent reviews of your work, and as you will see, the reviewers raised some concerns about the manuscript. Despite this, these reviewers are optimistic about your work and the potential impact it will have on research studying biogeography of flightless beetles. Thus, I encourage you to revise your manuscript accordingly, taking into account all of the concerns raised by the three reviewers.

Please note that reviewer 1 has kindly provided a marked-up version of your manuscript.

While the concerns of the reviewers are relatively minor, this is a major revision to ensure that the original reviewers have a chance to evaluate your responses to their concerns.

I look forward to seeing your revision, and thanks again for submitting your work to PeerJ.

Good luck with your revision,

-joe

Reviewer 1 ·

Basic reporting

Comments on Munoz-Tobar & Caterino
These comments are numbered as per the numbered comments on the ms.
1. The abstract mentions 4 species of Dercylus, yet the Introduction (line 91) lists 3. The numbers are complicated by the authors’ recognition of several undescribed species that were elucidated by the genetic analysis. You should go back through the manuscript and determine how best to list these species (e.g. “three described and two previously unknown, undescribed species”, or the like). Also, I have flagged various situations where there is conflict between the plurality of the subject and verb tense used. Also, lines 75-76 do not include a sentence. The final manuscript should be gone over for correct English usage.
2. The authors focus on genetic heterogeneity, however the geographic aspect of genetic variation is more important. Many species across a higher taxon have similar levels of heterogeneity. It’s how that heterogeneity is partitioned geographically that is information about popluational differentiation.
3. You should cite the original authors’ paper within which the Dercylus are described, i.e. include it in the refernces. It is only when taxonomists are given credit for their work that systematics will survive.
4. The fossils used to calibrate the Dyscolus phylogenetic analysis are not adequate. Instead of Larsson (1978) you should cite Schmidt (2015, Zootaxa 3974: 573-581) who actually described a fossil Limodromus (= Platynus) species dated as Eocene. Also, Scudder (1890, NOT 1980) was a horrible taxonomist, and often incorrectly assigned his fossil taxa to genera. Among the two “Agonum” from the Green River Formation, Platynus caesus looks to me to actually be a Clivina! His Platynus senex (p. 519, Plate 7, Fig. 38) could be an Agonum. I would have no problem with that assignment, especially given Liebherr and Schmidt (2004, Deutsche Entomologische Zeitschrift 51: 151-206) who assigned the origin of Agonum as Eocene based on cladistic biogeographic information (see Figs. 97, 105). Examination of the MCZ photo of P. senex--https://mczbase.mcz.harvard.edu/guid/MCZ:Ent:PALE-1844--suggests that the beetle could be a Europhilus like species. It is not like P. variolatus as Scudder suggests as the humeri are broadly rounded, but it can credibly serve as an Eocene placemarker in the plylogenetic analysis. Finally, the citation of Scudder is incomplete as well as incorrect as to year. I suspect this is an artifact of over-reliance on Endnote, which would have trouble with such a long title, volume number, special periodical, etc. As this reference is available via BHL, there is no excuse regarding how to properly cite it.
5. The neutrality tests for both genes have internally conflicting data. You state the populations are going through expansion, but a high proportion of rare alleles is not present. You cannot present both aspects. Better to delete these details. That is, skip mentioning whether the populations are expanding or not.
6. You mention individual SIMT208H6 which appears in the COI tree. But it is not in the CAD tree, where SIMT208H5 assumes the outgroup position. Also in 5A, SIMT208H5 is labeled SIMT028H5. The specimen numbers need to be gone over carefully to get rid of this sloppiness. And why exactly should SIMT208H6 not be assigned to Dyscolus alpinus in 5A, especially given the vagaries of mtDNA in accurately circumscribing species limits? AND FINALLY, why is Dyscolus spp. SIMT208H5 not considered a D. alpinus in 5B whereas it is considered so in 5A. This entire figure creates more problems than it solves, either through sloppiness or incorrect interpretation of the interaction of genetic data and taxonomy. You cannot just remove individuals from species willy nilly, and then put them back again when it suits.
7. There is much space utilized to present the individual COI and CAD trees for Dercylus, even their results are nonsensical. This is not surprising given shared ancestral polymorphisms in mtDNA. That the combined analysis results are credible means that only those should be presented. Delete Fig. 7 and its discussion.
8. 8. If you dump the COI tree for Dercylus this point of discussion disappears. Arguing that species are monophyletic or not based on mt DNA is indefensible.
9. Only in the discussion are we presented with information (unpublished) concerning Pelmatellus beetles. All mention of Pelmatellus results should be struck.
10. Fig. 4—if you are dealing with only 1 species, the abbreviation is “sp.” Also, Dercylus should be italicized in the CAD network.

I have looked at some of the documentation regarding permits, etc. and it seems genuine. However I view that as a role for the editor, not for a reviewer.

Experimental design

See above

Validity of the findings

See above.

Additional comments

This manuscript needs to be seriously streamlined, with only tree presentation of combined COI plus CAD data. Species limits must be carefully considered, rather than putting individuals into and out of species without taxonomic consequences. The entire manuscript should be carefully read for English usage after revision.

Annotated reviews are not available for download in order to protect the identity of reviewers who chose to remain anonymous.

Reviewer 2 ·

Basic reporting

I find this study well executed, of interest and well-written. I have only one major issue with the methods and otherwise a few comments:

I find the paper fairly long considering the amount of data that is generated, with a particularly lengthy discussion. Perhaps an additional effort to summarize the main ideas could be made to save some space.

Experimental design

Each genus should be presented in the introduction to give a sense of their respective species richness, knowledge of their phylogenetic placement and potential proximity, and monophyly of the species groups the authors are focusing on.

The choice of outgroups should also be better justified.

The dating exercise seems relatively simple but in fact is probably severely biased by the choice of the taxon sampling. The authors rely on fossils placed in the outgroups to calibrate their clocks. This would be acceptable if the taxon sampling was denser. As it stands, the taxon sampling does not allow a correct placement of the fossils and the ages are very likely to be incorrectly estimated. I would recommend that the authors repeat their analyses with a denser taxon sampling to allow a more accurate placement of the fossils in this dating scheme. I would also recommend that the authors compare their results with a clock calibration based on published rates of molecular evolution for the CO1 of Carabidae which are available from Andújar et al. (2012, BMC Evol. Biol.). Even if the rates can be different between their Andean beetles and the genus Carabus, this would give some sense of how reliable the divergence time estimates are. Also this can be accommodated to some extent using relaxed clocks as the authors used in this submission.

Validity of the findings

no comment

Reviewer 3 ·

Basic reporting

Line 75: Remove the word Where

Line 130: change to …one taxon was collected…

Line 347: From figure 6 it appears the D. alpinus clade originated 6.32 Mya with a confidence interval of 1 – 14 MYA not 5.1-8.0

Line 355: change to …each species represents a distinct clade…

Lines 372-376: include the 95% confidence intervals for the dates of Dercylus species

Figure 5.
- Make the font size of bootstrap and Bayes PP a little larger. (Also in Figure 8)
- Explain in the text or figure legend that only one sample of each haplotype was included in the phylogenetic analysis
- Explain in the figure legend what is meant by North, North West, Mixed, etc. or maybe use the site names instead. Make this more explicit in the text line 344-345. Put the focus on the site and the direction in parentheses.

Figure 9.
- I like the time labels for each species of Dercylus. Please also include a similar label for the genus in the Andes 30.13 (~12-47mya)

Experimental design

Line 133: IT would be good to include additional outgroups to the Dercylus phylogenetic and BEAST analysis as the deepest branches have low support. I did not see any additional Oodine or Licinine CAD sequences in GenBank, but there are a few CAD sequences in the Harpalinae subfamily that might be useful and there may be many COI sequences that would be useful in GenBank.

Line 159-194: Please include (probably as supplemental information) the details of the settings and parameters for each of the analyses for MrBayes, RAxML, DNAsp, TCS, BEAST, Arlequin, and Structure so one can replicate your results. If you used only defaults, please say so.

Validity of the findings

Line 96 and 404: Aim of the study: I felt uncomfortable with the comparison of these to taxa with regard to how flightlessness has affected their phylogenetic structure and population connectivity. I would not expect – a priori- so see similar patterns of genetic diversity and gene flow when comparing populations of a single species (D. alpinus) to species of a genus (Dercylus), especially if they are different ages. There is more going on here than just flightlessness. The history of these taxa is very different. Make it clear that you are comparing apples and oranges. It is not surprising that there is very little gene flow between different species that have been separated for a long time and that there is more gene flow or evidence of recent gene flow among populations of the same species that have been separated relatively recently. I think you should put more emphasis on how these are two different scenarios and not try to connect them so much with flightlessness.

---

## Round 0.2 · accepted · Accept

· Academic Editor

Accept

Dear Drs. Muñoz-Tobar and Caterino:

Thanks for re-submitting your manuscript to PeerJ, and for addressing the concerns raised by the reviewers. I have checked your revisions and now believe that your manuscript is suitable for publication. Congratulations! I look forward to seeing this work in print, and I anticipate it being an important resource for research on the biogeography of flightless beetles. Thanks again for choosing PeerJ to publish such important work.

-joe